# Investigation of the Friction Properties of a New Artificial Imitation Cartilage Material: PHEMA/Glycerol Gel

**DOI:** 10.3390/ma16114023

**Published:** 2023-05-28

**Authors:** Zikai Hua, Mindie Hu, Yiwen Chen, Xiuling Huang, Leiming Gao

**Affiliations:** 1School of Mechatronic Engineering and Automation, Shanghai University, Shanghai 200444, China; 2Department of Engineering, Nottingham Trent University, Nottingham NG1 4FQ, UK

**Keywords:** synthetic cartilage, poly(hydroxyethyl methacrylate) (PHEMA), synthetic gel, compression, coefficient of friction, wear

## Abstract

The absence of artificial articular cartilage could cause the failure of artificial joints due to excessive material wear. There has been limited research on alternative materials for articular cartilage in joint prostheses, with few reducing the friction coefficient of artificial cartilage prostheses to the range of the natural cartilage friction coefficient (0.001–0.03). This work aimed to obtain and characterize mechanically and tribologically a new gel for potential application in articular replacement. Therefore, poly(hydroxyethyl methacrylate) (PHEMA)/glycerol synthetic gel was developed as a new type of artificial joint cartilage with a low friction coefficient, especially in calf serum. This glycerol material was developed via mixing HEMA and glycerin at a mass ratio of 1:1. The mechanical properties were studied, and it was found that the hardness of the synthetic gel was close to that of natural cartilage. The tribological performance of the synthetic gel was investigated using a reciprocating ball-on-plate rig. The ball samples were made of a cobalt-chromium-molybdenum (Co-Cr-Mo) alloy, and the plates were synthetic glycerol gel and two additional materials for comparison, which were ultra-high molecular polyethylene (UHMWPE) and 316L stainless steel. It was found that synthetic gel exhibited the lowest friction coefficient in both calf serum (0.018) and deionized water (0.039) compared to the other two conventional materials for knee prostheses. The surface roughness of the gel was found to be 4–5 μm through morphological analysis of wear. This newly proposed material provided a possible solution as a type of cartilage composite coating with hardness and tribological performance close to the nature of use in wear couples with artificial joints.

## 1. Introduction

Osteoarthritis has a high incidence in the elderly population [1], and it is often accompanied by severe pain or even an inability to complete daily activities [2]. The current treatment of advanced joint disease to restore joint physiological function is artificial joint replacement surgery. Artificial joints consist of a Co-Cr-Mo alloy or ceramic ball paired with a UHMWPE cup and are commonly used during treatment. However, artificial articular cartilage is still lacking in the frictional pair. Over the course of medium- and long-term usage, these prosthetic materials inevitably produce particles due to wear [3]. For the plastic materials, these wear particles could cause osteolysis (bone resorption), which leads to aseptic loosening and a final failure of the prosthesis. This is one of the main reasons for the revision of prostheses currently [4,5,6]. For metal materials, the released elemental ions, such as Co or Cr, could immigrate into a patient’s circulatory system and even cause an allergic reaction [7,8]; moreover, the biotribocorrosion of metal still exists in the corrosive environment of the human body even with the development of surface modification [9]. Ceramic materials could affect the life quality of patients due to the abnormal noise caused by friction [10,11]. Hua et al. found that squeaking is more likely to occur when there is a higher friction coefficient at the contact area of ceramic-on-ceramic hip prostheses [12]; it was speculated that the addition of artificial cartilage could buffer resonance. In order to supplement the lack of articular cartilage in current artificial prostheses and address the friction problem between joint-bearing surfaces, research on artificial synthetic cartilage materials is of particular importance. The gel, as a highly flexible polymer network filled with three-dimensional structures of polymer chains by liquid macromolecules, could be an advanced material for artificial imitation cartilage. Previous research has shown the efficiency of hydrogel-based therapy in correcting and repairing knee cartilage defects, while hydrogel-based treatments can also improve pain and functional scores in patients with knee defects [13]. The application of soft gel materials on the artificial cartilage of human joints has been studied in terms of their mechanical and tribological performance [14,15,16]. Polyvinyl alcohol (PVA) hydrogel has been applied to the replacement of joint prostheses, and Stoimenov et al. obtained the frictional coefficient in metal (<0.05–0.08) and in polymers (0.4–1.5) [17]. The friction coefficient of a new type of agose-hydrogenan hydrogen, which can reinforce the three-dimensional polymer network and has good wear performance, was 0.05–0.35 [18]. PVA–hydroxyapatite (HA) and PVA–HA–poly(acrylic acid) (PAA) composite hydrogels are another new substitute for articular cartilage, with a frictional coefficient of 0.035 ± 0.0028 [19]. There is a certain gap in the coefficient of friction with natural cartilage (0.001–0.03). Therefore, the mechanical and tribological performances of gel materials when used for artificial articular cartilage in artificial joints need further investigation for the improvement of tribological performance in order to imitate natural articulate cartilage. We aimed to research a new gel material and study its mechanical and tribological performances to verify its potential as a substitute for articular cartilage in joint prostheses.

PHEMA has long been used to make contact lenses [20], which proves that it is a good biomedical material in terms of biocompatible and hydrophilic properties. However, it is hard with brittle properties and poor elasticity, which has limited its use as the matrix of cartilage material. Many researchers have introduced water or other media in order to reduce the hardness and make it a gel-like elastomer. Passos et al. proved that the thermal stability, tolerance to organic solvents, and cytotoxicity of pure PHEMA hydrogel could meet the requirements of artificial cartilage in tissue engineering [21]. To improve its performance, various chemical components have been introduced. For example, Bavaresco et al. blended HEMA with n-vinyl pyrrolidone (NVP) to prepare the gel [22]; Bostan et al. used HEMA and acrylic acid (AA) to copolymerize [14]. However, the biocompatibility of these components blended in PHEMA copolymerization remains to be further investigated. With regard to the tribological properties of synthetic gels, there have been few reports on friction tests between artificial cartilage and joint prosthesis materials.

In this work, we used HEMA as the matrix and glycerol as the hardness regulator and then polymerized it in a water bath. Glycerol was selected as a gel polymerization regulator because it is a natural metabolite of the human body with good biocompatibility and is also a good lubricating candidate for joint frictional pairs. In our previous research, the optimal mass ratio of the HEMA monomer to glycerol was found to be 1:1 [14,23]. The obtained gel had the best compression performance and elasticity compared to other mass ratios. It also showed low surface roughness, and its hardness was close to natural. It is expected that the lubrication of current artificial joints and the motion range of joint prostheses could improve with this new synthetic cartilage material [24]. Moreover, it also provides the feasibility of the clinical application of cartilage tissue engineering.

The friction coefficients, wear surface profiles and surface roughness were obtained in the results. It was found that the gel exhibited the lowest friction coefficient among the other two materials, and the wear resistance performance was improved since no metal ions were observed from the metal-on-gel pair.

## 2. Materials and Method

### 2.1. Preparation of Gel

The raw materials and reagents for the preparation of artificial cartilage prostheses are shown in Table 1.

HEMA and glycerin were injected into an Aluminum foil crucible, then mixed at a mass ratio of 1:1. After that, 0.2 wt% initiator ammonium persulfate was added to the foil crucible. As a preparation process for stirring, a magnetic rotor was put into the mixed solution. This crucible was placed in a magnetic stirrer with a constant temperature water bath of 70 °C, and the crucible was uniformly stirred to prepolymerise for about 15 min. When the solution was slightly viscous, the magnetic rotor was taken out, and the temperature in a constant temperature water bath was kept at 70 °C for 24 h until the reaction was complete. The flow chart for the preparation is shown in Figure 1. After washing the artificial imitation cartilage material with deionized water for 30 min to remove the residual initiator of the sample, the material was placed in a vacuum drying chamber and dried for 24 h. The UHMWPE plate and the stainless-steel plate were cleaned in the ultrasonic cleaner, and the plates were dried with a jet of filtered inert gas, referring to the cleaning methods of ISO14242-2 [25]. All procedures of tests have been conducted in the bio-tribological laboratory at the relevant humidity of 40% and a temperature of 37 ± 2 °C.

### 2.2. Compressive Elasticity Test

Articular cartilage plays a dominant role in cushioning motion pressure in the human body; therefore, it is important to determine the compressive properties and elasticity of this new material [26]. The evaluated hardness of the gel material was about 13% less than the porcine cartilage in another study [23]. A 2 mm twist drill was used as the measuring part instead of the needle part of the penetrometer, and we measured the drill’s penetration, which represented the hardness of the gel material. The samples prepared in the method described above were tested for compression resistance and elastic recovery using a tensile and compression tester. The samples were prepared in a 30 × 50 mm rectangular mold, and the entire surface of the sample with a surface area of 1.5 × 10^−3^ m^2^ was subject to a compressive force. The axial pressure within the equivalent force range of 600 N to 1000 N was applied to the sample with an increment of 100 N and was controlled by a displacement speed of 2 mm/min. When each loading step was reached, the pressure was kept for 3 min and unloaded/rested for 3 min, as shown in Figure 2. After each loading step, the thickness of the sample was measured.

### 2.3. Tribological Performance Test

The tribological properties of the gel in sliding were investigated using a reciprocating ball-on-plate friction tester (Sliding-POD, Orthotek Lab, Shanghai, China). The balls were made of a Co-Cr-Mo alloy, and the plates were made of synthetic PHEMA/glycerol gel, UHMWPE, and 316L stainless steel, respectively. Although the CoCr-stainless steel pairs are not currently used in clinics, we added this metal–metal friction pair as the control group mainly because we were interested in the changing trends of such contact pairs. The most used metal–metal materials in clinics are CoCr-on-CoCr, and one can find their frictional performance in the literature [3,17]. Calf serum and water were employed as the lubricating medium for human joint wear simulation. The PHEMA gel plates for the friction test were obtained by pouring the prepared well-mixed solvent into a prepared 30 × 50 mm rectangular mold with UHMWPE at the bottom of the mold. A uniform film of 2 mm in thickness was coated over UHMWPE, as shown in Figure 3.

The reciprocating friction and wear tester is shown in Figure 4 and Figure 5. The non-friction contact area of the sample was perforated and screwed to the test platform. The PHEMA gel plate against a 28 mm diameter Co-Cr-Mo ball was tested as the frictional pair. For comparison, the UHMWPE plate and 316L stainless steel plate of the same size, which are commonly used as hip prosthesis materials, were tested. The surface roughness (Ra) of the Co-Cr-Mo ball was less than 0.05 μm, according to the hip prosthesis standard ISO 7206-2 [27].

In earlier studies, the range of natural normal joint contact stresses has been proposed as 0.1–2.0 MPa [28,29]. In this study, the contact area between the ball and the surface of the plate was acquired with the use of pressure-sensitive films so that the contact pressure could be calculated. The average contact pressures of the three pairs, Co-Cr-Mo to gel and Co-Cr-Mo to UHMWPE, Co-Cr-Mo to 316L stainless steel, were controlled at 0.480 MPa, 0.513 MPa, and 0.634 MPa, respectively, reflecting the different stiffness of each material. The filtered calf serum was selected as a lubricating medium according to the wearing test standard ASTM F732-17 for artificial prosthesis materials, and deionized water was used as the control group [30]. A long-term wear test of 43,000 sliding motions was carried out with a frequency of 2 Hz, a sliding speed of 30 mm/s, a movement amplitude of 7.5 mm, and a total time of 6 h. After each hour, the frictional and normal forces were continuously recorded for 10 min and used to calculate the friction coefficient, reflecting the general trend within the total six hours. They were then represented with an average value for the overall average friction coefficient.

Since the synthetic gel was soft and easily deformed after compression, there was no pre-compression at the start of testing to ensure that the friction occurred at its original shape.

The morphology of the samples was obtained by the 3D Optical Surface Metrology System (Leica DCM-8, Leica Microsystems, Wetzlar, Germany) to observe the worn surface modification during the test. By the end of the experiment, the lubricating medium was collected, and the Cr and Mo elements in the deionized water were analyzed by an M-type oil spectrometer to evaluate the wear results during the whole experiment.

## 3. Results

### 3.1. Analysis of Compressive Elasticity Test

After each loading step represented in Section 2.2, the thickness of the sample was measured, and the elastic recovery ability of the sample was analyzed, as shown in Table 2. Under the loading range of 600–1000 N, the deformed gel was stable at 58–67% of the original thickness. After compression, no cracking phenomenon was observed in the gel plate. After unloading for 3 min, the gel plate gradually recovered to 97% of its original thickness, which demonstrated its good elasticity.

### 3.2. Comparative Analysis of the Friction Coefficients

The variations in the friction coefficients during the testing time for the three materials are shown in Figure 6. The initial friction coefficient of the gel plate was very low, which was about 0.0085, and finally reached 0.018 and 0.043 in the calf serum and deionized water, respectively. Despite different lubricating mediums, it gradually increased with an increase in the testing time and eventually tended to be stable at about 4 h.

The friction coefficients of the UHMWPE plate and 316L stainless steel plate decreased with an increase in the friction time, contrary to the gel plate, and this change did not depend on the type of lubricating medium either. Among them, the frictional pair Co-Cr-Mo ball to the 316L stainless steel plate in the deionized water showed the largest decrease in friction coefficients by 37.9% from 0.178 to 0.110. For all these cases, the variation in the friction coefficients during friction time was sharper in the calf serum than in the deionized water.

The overall-averaged friction coefficients for the 6 h testing were compared in Figure 7. The gel plate exhibited the lowest coefficient of friction in either the calf serum (0.018) or in the deionized water (0.039) compared to the UHMWPE or 316L stainless steel groups. The overall-averaged friction coefficients of the gel group were only 26.1–35.9% compared to the other two groups in the calf serum and 26.4–53.5% in the deionized water.

### 3.3. Morphological Analysis of Wear

The surface roughness of the plate samples before and after friction testing was extracted from the 3D micrographs, as shown in Figure 8. The surface roughness was found to be increased slightly after testing for most cases. It was found to decrease only in the case of the UHMWPE plate in deionized water, which was also observed in the morphology images, which showed how the initial surface processing traces disappeared.

The morphology images of the initial plate surfaces (before the test) are shown in Figure 9, and the worn surfaces are shown in Figure 10, Figure 11 and Figure 12 for the three different materials, respectively. The sliding direction was horizontally parallel to the images. A rougher surface of the gel plate could be observed in Figure 9a compared to the other materials, and these processing traces could be clearly seen on the UHMWPE and the steel plates in Figure 9b,c.

After friction testing, the gel plate maintained a relatively complete surface structure in the calf serum, with only a few scratches along the sliding direction, as shown in Figure 10. While in deionized water, a more distinct change in the surface roughness with more scratches could be observed. In the case of the UHMWPE plate, there were obvious friction marks on the surface of the calf serum, and the machined marks became shallow, as shown in Figure 11a. The friction traces were severer in deionized water because part of the initial surface processing traces even disappeared, as shown in Figure 11b. In the case of the 316L stainless steel plate, the friction marks were so obvious that even some surface processed traces disappeared, as shown in Figure 12.

### 3.4. Wear Element Released in Lubricating Media

The ion levels of Cr and Mo collected in lubricants after wear testing are shown in Table 3. Some Cr ions were found in the metal-on-316L stainless steel pair, which was obviously released from the metal ball, whereas no metal ions were found to be released from the metal-on-gel frictional pair.

## 4. Discussion

### 4.1. Compressive Properties

It is known that in human daily activities, the articular cartilage always bears compressive stress, and such a mechanical characteristic is important for the gel’s application as an articular cartilage biomaterial [31]. The mechanical properties of the PHEMA/glycerol gel depend on the content ratio of glycerol. It has been proven that the mass ratio of HEMA to glycerol at 1:1 is optimal in terms of elastic recovery and the compression resistance experienced by the compression test [22]. The gel plate could recover to 97% of its original thickness and be subjected to up to a 1000 N load without a surface crack observed; it exhibited viscoelastic behavior. Moreover, the hardness of the synthetic gel was close to the natural cartilage [32], and the residual stress after unloading could be avoided, which could benefit the cartilage restoration.

The frictional properties of gels were also partly determined by their mechanical properties [33]. The excellent compressive properties that the gel exhibited contributed to its low friction coefficient.

### 4.2. Coefficient of Friction

An obviously different trend in the friction coefficient variations was found that the coefficient was decreased during testing in the metal-on-gel pair while the other two material pairs showed an increasing trend during this time (Figure 6). These variations were observed to be independent of the types of lubricating medium. The possible reason for this was that the exchange between water and glycerol in the porous gel material changed its mechanical properties during the long-time friction. In the manufacturing process of the gel material, no water was added. When the gel was immersed in the lubricating medium, small water molecules infiltrated into the gel’s porous structure, and the gel material gradually absorbed the water. With the continuous friction and loading applied, the surface layer of the gel was extruded and rubbed so that glycerol was squeezed out of the surface, and water molecules replaced some glycerin gradually at the surface layer.

Firstly, the unpolymerized glycerol components were observed on the gel surface, and they played a role in lubrication, leading to lower initial friction. However, after some time, the glycerin was dissolved in the lubricating medium, and the benefit no longer existed. Secondly, the mass ratio of HEMA and glycerin did not reach the optimal rate of 1:1 anymore, and the change in the glycerin and water content could have reduced the mechanical strength of the gel material. As discussed in the compression test, the elasticity and hardness of the synthetic gel could be affected. Thirdly, the change in the gel components could affect the viscoelastic property of the gel. After a long-time, the continuous reciprocating of sliding the material became more viscous, resulting in larger deformation in the shear direction. This deformation increased the shear stress and contributed to an extra frictional force. The upward trend in the friction coefficient in the current gel materials was in line with the friction test of natural cartilage [34]. It suggested the synthetic gel material behaves similarly to the natural cartilage in terms of the variations in the friction coefficients.

For all the tested cases, a common trend was that the friction coefficients gradually reached a nearly stable value in about 4 h. It suggested that with continuing friction, the structure of the gel composition gradually reached a steady-state stage, as did the other two material pairs. The synthetic PHEMA gel plate showed the lowest friction coefficient (roughly in the range of 0.01–0.04, as shown in Figure 7) among the three material pairs tested. It demonstrated that the operating condition was in the mixed lubrication regime according to the typical Stribeck curve of lubrication. The stable friction coefficient for the synthetic gel pair in the calf serum was about 0.018. In previous works in the literature, the friction coefficient of natural cartilage was in a range of 0.001–0.03 [35,36]. The friction coefficient of the polyvinyl alcohol (PVA) hydrogen, which was applied to artificial cartilage materials, was about 0.04–0.147 [17,37]. The friction coefficient of a new type of agose-hydrogenan hydrogen was 0.05–0.35, and the well-wear performance was owed to reinforcing the three-dimensional polymer network [18]. The friction coefficient of PVA–hydroxyapatite (HA) and PVA–HA–Poly(acrylic acid) (PAA) composite hydrogels was 0.035 ± 0.0028 [19]. Bavaresco et al. studied the complex effect of the reinforcing polymer and crosslinking density on the tribological properties of PHEMA-based hydrogels (PHEMA/poly(MMA-co-AA) hydrogels and PHEMA/NVP hydrogels) when sliding against stainless steel 316L in distilled water, and the friction coefficient values were relatively low (0.01–0.03) [22]. The friction coefficient of the gel in this study had the same order of magnitude as the above artificial hydrogen materials, but it also had a relatively lower value and was within the range of natural cartilage.

Regarding the lubricating medium, it was observed that the calf serum provided better lubrication for all frictional pairs compared to the deionized water in all cases, and the friction coefficients were more stable during the 6 h test (Figure 6). This was because the fluid properties of calf serum were closer to human synovial fluid with a higher viscosity containing proteins so that it could support better lubricating the film in the mixed lubrication regime of the Stribeck curve. The presence of biomacromolecules, such as proteins in calf serum, balanced the concentration of the liquid inside and outside the gel and reduced the loss of glycerol. Some studies have shown a similar impact of lubricants on the tribological properties of the hydrogel as an artificial cartilage, and the coefficient of friction tested in deionized water was twice tested in albumin [38,39]. The importance of albumin and bovine serum in lubrication has been studied by film thickness measurements conducted in ball-on-disc testing; the aggregation and absorbance of the protein occurred at the surface with a thin protein film to help reduce friction [40].

### 4.3. Surface Morphology

The surface roughness of the gel slightly increased during the test. Li et al. found polyvinyl alcohol hydrogel to be artificial cartilage material, and the observed surface roughness increased slightly after wear in both albumin solution and deionized water [38]. Surface asperity and its plastic deformation are crucial aspects to consider, as the asperity interactions between the two substrates considerably influence friction [37]. The surface roughness of the tested samples was not optimum in the current study, 1–3 μm for the UHMWPE and stainless stall plates and 4–5 μm for the gel plates, as shown in Figure 8, which could be further reduced to around 0.86 μm when closer to the surface roughness of natural cartilage, and the friction coefficient for all pairs could be reduced as well [41].

It was observed that a large area of the original machining traces disappeared in the 316 stainless steel plate and UHMWPE plate, with a maximum of a 20 μm depth of friction scratches, in both the calf serum and the deionized water. It seemed that the surfaces were polished during friction and resulting in reduced friction coefficients.

There were no metal ions observed from the metal-on-gel pair, which suggested a better wear resistance performance compared to the other soft (UHMWPE)-on-hard material pair. Avoiding the release of metal ions from the metal ball component was a great benefit of reducing adverse wear particle reactions with human joint tissues.

The preliminary tribological study also had some limitations. Ammonium persulfate, as a toxic material, can be carefully considered as an initiator for the preparation of gel. Due to the short duration of testing, no gravimetric measurements of wear were taken during this work. On the other hand, the swing and torsion friction study of the hydrogel in terms of the flexion-extension and adduction-abduction of the hip and knee joints were found, and the sliding and rotation speed had a great influence on the friction behavior of the hydrogel study [42]. The randomness of the joint motion direction in human activities was proved to affect the wear factor and needed to be taken into consideration [43]. The friction coefficient between the natural articular cartilage and hydrogel was also found to significantly depend on the load [37]. These factors should be examined in the long-term test to determine how multi-direction, speed, load, randomness and lubrication can infect the tribology of the gel. Moreover, the surface roughness of the gel after testing could be reduced to 0.86 μm, which is also the key to further reducing its friction coefficient. To give a particular assessment of the compressive property of the gel, a long-term compressive force could be applied to the gel. The component of the gel changed as the water molecules were infiltered into the porous structured material; therefore, more detailed work on the component’s numerical record is expected in future work.

The gel can only function as a cartilage replacement once sufficiently integrated with the implant or bone surface [44]. In previous research, gels as softer materials were designed with adhesive properties, using tissue adhesives or sutures and ensuring a secure fit [45,46]. The fixation method for the implanting of the gel–cartilage interface should be considered and evaluated in future work.

## 5. Conclusions

In this work, a new artificial joint cartilage material, PHEMA/glycerol gel, was proposed and tested with mechanical compression and friction simulations, introducing the reciprocating friction and wear tester. The synthetic gel material was coated on the UHMWPE plate, sliding against a Co-Cr-Mo ball. Another two materials for the plates, UHMWPE and 316L stainless steel were tested for comparison. The conclusions are drawn below.

The synthetic gel with HEMA and glycerin at a mass ratio of 1:1 exhibited excellent compressive properties in terms of elastic recovery and compression resistance.The PHEMA/glycerol gel showed a low friction coefficient which was close to the natural cartilage, and no metal ion was released in the lubricant.The friction coefficient increased during testing, probably due to the gel’s poro- and visco-elastic properties, and reached a stable structure in about 4 h.

The proposed PHEMA gel material was tested and was proved to effectively reduce the friction of the artificial joint prosthesis, indicating its excellent lubrication performance. Moreover, the biological safety of this synthetic gel was confirmed to ensure that patients had little chance of suffering from osteoarthritis and allergic reactions due to wear products since no metal ions were observed through surface spectroscopic analysis after the wearing test. This new material could, therefore, be used as a cartilage composite coating for artificial joints, which could be a possible solution to reduce the rising temperature and noise, and prolong the service life of the joint prosthesis, particularly for hip and knee joints. Our results provide a new option as a potential articular cartilage substitute in clinical applications for cartilage tissue engineering due to the good porous elastic property, compression resistance, frictional property and biological safety of this synthetic PHEMA gel material.

## Figures and Tables

**Figure 1 materials-16-04023-f001:**
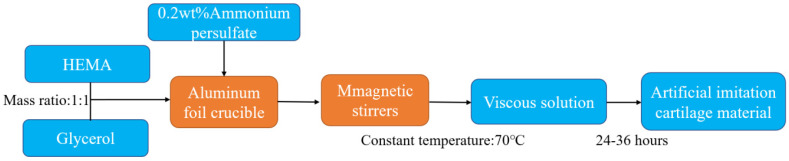
The process of preparation.

**Figure 2 materials-16-04023-f002:**
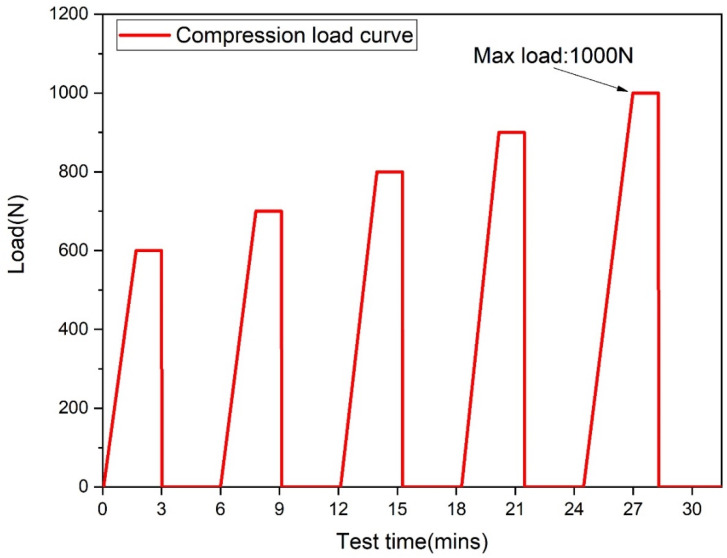
Test condition in the compressive test.

**Figure 3 materials-16-04023-f003:**
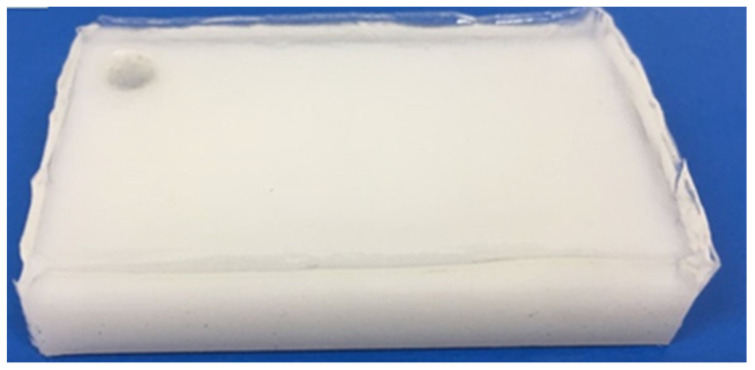
Preparation of gel sample.

**Figure 4 materials-16-04023-f004:**
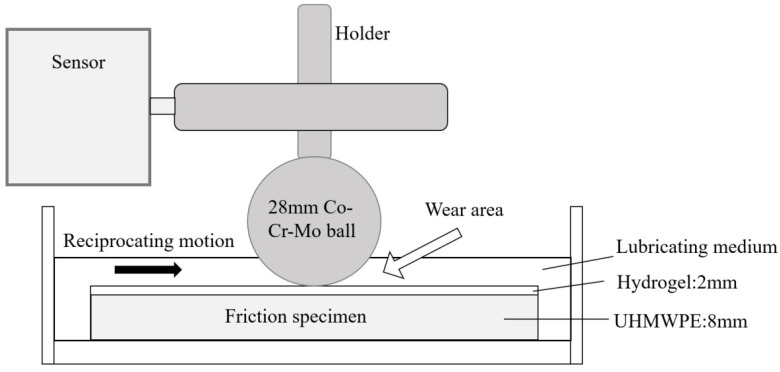
Installation of gel sample.

**Figure 5 materials-16-04023-f005:**
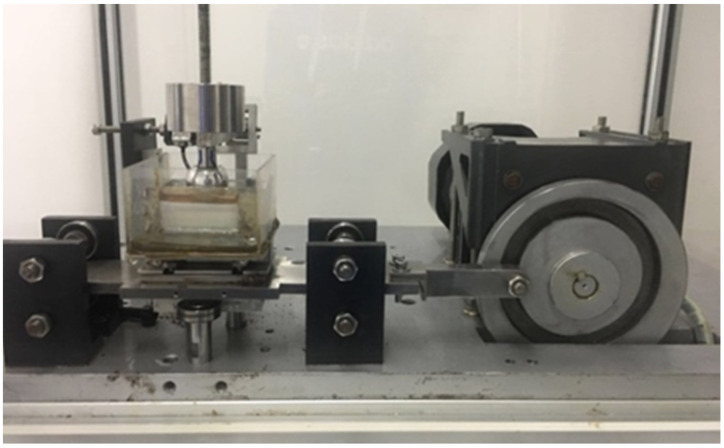
Reciprocating friction and wear testing machine.

**Figure 6 materials-16-04023-f006:**
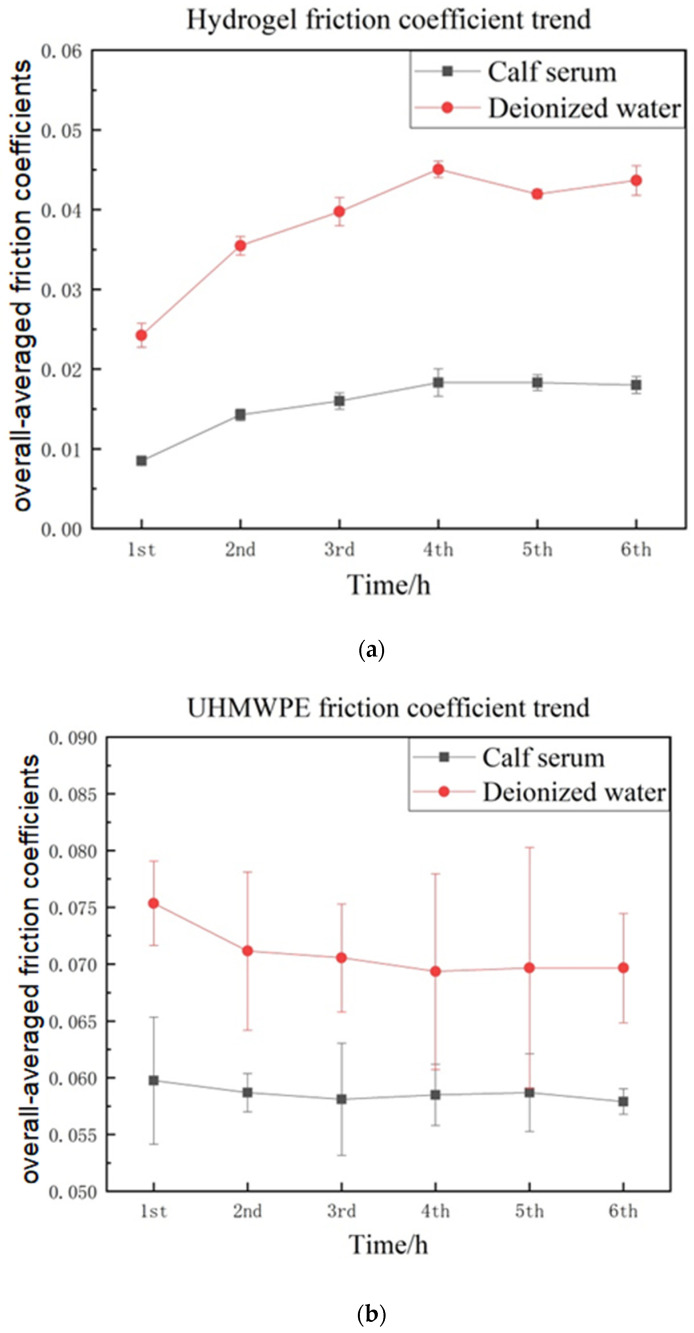
Friction coefficient of the Co-Cr-Mo ball against the plate made of (**a**) Synthetic PHEMA/glycerol gel (**b**) UHMWPE (**c**) 316L stainless steel.

**Figure 7 materials-16-04023-f007:**
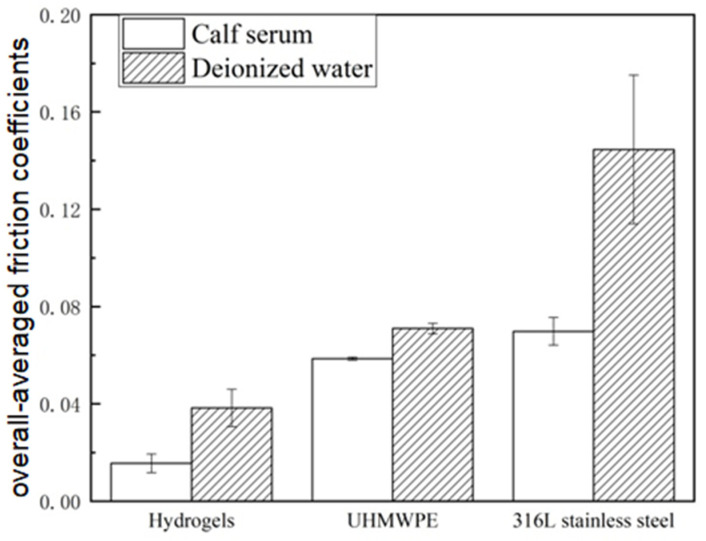
Comparison of averaged friction coefficient between frictional pairs.

**Figure 8 materials-16-04023-f008:**
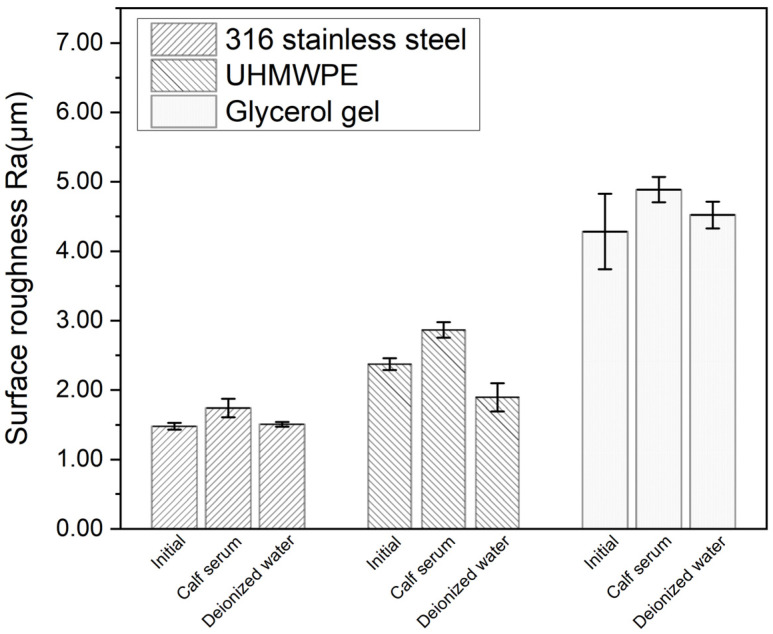
Comparison of surface roughness between frictional pairs.

**Figure 9 materials-16-04023-f009:**
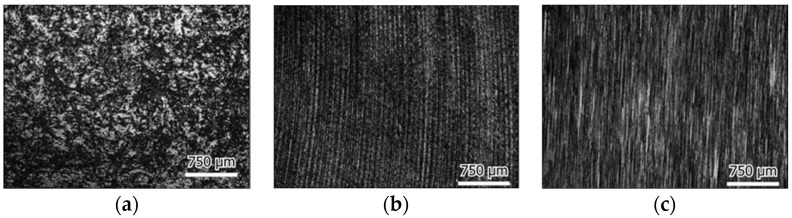
Initial surface of (**a**) Gel, (**b**) UHMWPE and (**c**) 316L stainless steel plate.

**Figure 10 materials-16-04023-f010:**
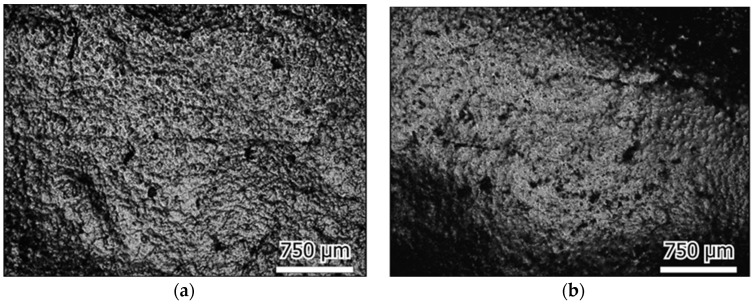
PHEMA/glycerol gel wear surface; sliding in the horizontal direction in: (**a**) Calf serum (**b**) Deionized water.

**Figure 11 materials-16-04023-f011:**
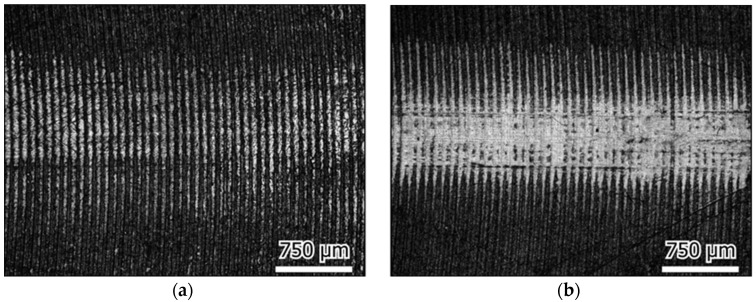
UHMWPE plate wear surface; sliding in the horizontal direction in: (**a**) Calf serum (**b**) Deionized water.

**Figure 12 materials-16-04023-f012:**
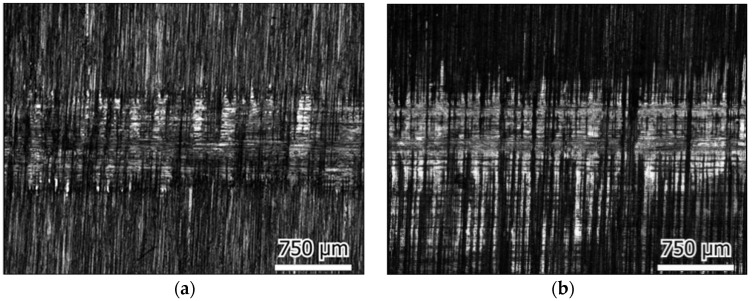
316L stainless steel plate wear surface; sliding in the horizontal direction in: (**a**) Calf serum (**b**) Deionized water.

**Table 1 materials-16-04023-t001:** Materials for gel preparation.

Material	Supplier
Hydroxyethyl methacrylate (HEMA)	AR, Chengdu Micxy Chemical Co., Ltd., Chengdu, China
Glycerol (medical glycerin)	Shangqiu Liangfeng Hygiene Products Co., Ltd., Shangqiu, China
Ammonium persulfate ((NH_4_)_2_S_2_O_8_)	AR, Chengdu Jinshan Chemical Reagent Co., Ltd., Chengdu, China
Calf serum	Gibco Cell Culture, Shanghai, China

**Table 2 materials-16-04023-t002:** Result of compressive elasticity test.

Load (N)	Percentage of Original Thickness Under Load (%)	Percentage of Recovery after 3 Minutes of Unloading (%)
600	67	100
700	64	100
800	63	99
900	62	98
1000	58	97

**Table 3 materials-16-04023-t003:** Spectral analysis results of wear elements/ppm.

Plate Material	Cr	Mo
PHEMA/glycerol Gel	0	0
UHMWPE	0	0
316L stainless steel	2.57	0

## Data Availability

The data presented in this study are available on request from the corresponding author.

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
