# Peer review of "Investigation of the Friction Properties of a New Artificial Imitation Cartilage Material: PHEMA/Glycerol Gel"

_materials, 2023, doi:10.3390/ma16114023_

Round 1

Reviewer 1 Report

Comments and Suggestions for Authors

The manuscript entitled “Investigation on the friction properties of a new artificial imitation cartilage material: PHEMA/glycerol gel “. I recommend major revisions because of:

1.      The abstract does not highlight the relevance of the study and does not present its novelties. One would also expect some relevant values obtained as results from the research to be mentioned.

2.    Introduction: it is too poor in content and citations to support the relevance and novelty of the research. The citations are not connected to the cases being studied and do not provide relevant information that highlights why what is being studied is important.

3. The objective, methodology, and results should be better described, discussed and

justified.

4. The results should be expanded significantly and quantitatively

5. I strongly suggest that authors shall carry out more studies to compare the results

from this paper to that from other similar studies.

Need English editing

Reviewer 2 Report

The results of the investigation of new gel material that can be used as a component of the artificial cartilages are undoubtedly important as one of the approaches to improve the quality of joints prostheses.

The manuscript presented can be of interest to the readers of the Journal.

But there are some questions and notes to arise while reading the manuscript.

1. The authors should clarify the real concentration of glycerol in the material under tests at different stages of its use. It is stressed in the text that “The mechanical properties of the PHEMA/glycerol gel depend on the content ratio of glycerol”. In the “Materials and methods” section, it is reported that initially HEMA was mixed with glycerol in a ratio of 1:1. However, then the composition of the material very likely changed already in the process of washing in an aqueous medium. It is known that glycerol is highly miscible with water, so that in an aqueous environment it will obviously be extracted out of the gel. So it is necessary to present the results of the analysis confirming the final concentration of glycerol in the material before testing, as well as to characterize the dynamics of changes in the concentration of glycerol in the material during tribological tests, which are carried out in water medium for 6 hours. In the article, the authors discuss the possibility of changing the composition of the gel, primarily in the near-surface layer, in the process of friction in water. But no specific data characterizing this process can be found in the article. It is advisable to obtain quantitative characteristics of this process.

2. The information concerning the dimensions of the sample tested in compression mode, the area of the material on which the pressure was applied, should be presented. Without this data, information about the magnitude of the load applied to the sample (600-1000 N) is meaningless.

3. The description of the results of compression tests should be transferred to the “Results” section.

4. In the working cycle in the joint prosthesis, the hydrogel will be subjected not only to impulse compression acts, but also to prolonged exposure to compressive force. The authors stress that “It is known that in human daily activities, articular cartilage always bears compressive stress, and such mechanical characteristic is important for gel’s application as an articular cartilage biomaterial”. Therefore, along with experiments on the short-term application of a compressive force, it is very important to test the material in the long-term creep regime and present the appropriate results in the article – first of all the “time-deformation” curves.

5. What is the type of the friction tester used? The producer of the equipment should be presented.

6. The amplitude if the movement of the counterbody during the friction test should be included in the description of this experiment along with the overall run during 43,000 sliding motions.

7. The authors use the term "hardness" several times, for example: "…the hardness of the synthetic gel was close to the natural cartilage". It should be explained to readers what namely mechanical characteristic the authors have in mind.

8. The authors stress in the “Discussion” section that to obtain the gel material with the optimal friction characteristics one should reduce the surface roughness: “The surface roughness of tested samples was not optimum in the current study, 1-3 mcm for the UHMWPE and stainless stall plates and 4-5 mcm for the gel plates as shown Fig. 8, which could be further reduced to the order of magnitude of 0.01-0.1 mcm and the friction coefficient for all pairs could be reduced as well”. But it is well known that the surface of real articular cartilage is not smooth. It is characterized by a rather deep microrelief, which just provides the optimal conditions of lubrication of the friction surface during the operation of the joint. Apparently, the authors should think about using this possibility of optimizing the friction process.

9. The authors should carry out a deep revision of the quality of the English language in which the article is written.

See the comments to the authors, please.

Round 2

Reviewer 1 Report

Dear editor

The authors replied to all comments. 

Best regards